# Combined Static Stretching and Electrical Muscle Stimulation Induce Greater Changes in Range of Motion, Passive Torque, and Tendon Displacement Compared with Static Stretching

**DOI:** 10.3390/sports11010010

**Published:** 2023-01-04

**Authors:** Takamasa Mizuno

**Affiliations:** Research Center of Health, Physical Fitness and Sports, Nagoya University, Furo-cho, Chikusa-ku, Nagoya 464-8601, Japan; mizuno@htc.nagoya-u.ac.jp; Tel.: +81-52-789-3959

**Keywords:** joint flexibility, stiffness, ultrasonography, passive torque, plantar flexors

## Abstract

The purpose of this study was to determine the combined effects of static stretching and electrical muscle stimulation on maximal dorsiflexion angle and passive properties. Sixteen healthy subjects participated in three randomly ordered experimental trials: combined static stretching and electrical muscle stimulation, static stretching alone, and control. In combined trial, subjects performed 5 min of calf stretching while receiving electrical muscle stimulation of the gastrocnemius medialis. In static stretching trial, subjects performed calf stretching only. Maximal dorsiflexion angle, passive torque, and muscle displacement were measured before and after intervention. Tendon displacement was also calculated. The difference from pre- to post-intervention in maximal dorsiflexion angle in combined trial was greater compared with that in the control (*p* = 0.026), but the static stretching trial exhibited no significant difference (both *p* > 0.05). Passive torque at submaximal dorsiflexion angles was significantly decreased only after combined trial (all *p* < 0.05). Muscle displacement at maximal dorsiflexion angle was significantly increased in all conditions (all *p* < 0.05). Tendon displacement at maximal dorsiflexion angle was higher after combined trial compared with static stretching trial (*p* = 0.030). These results revealed additional effects of adding electrical muscle stimulation to static stretching on maximal dorsiflexion angle, passive torque, and tendon displacement.

## 1. Introduction

Some recent guidelines state that flexibility training has a beneficial role in improving and maintaining the physical fitness and health of adults [1,2]. Static stretching (SS) is a common method for increasing joint flexibility and is widely used as an integral part of warm-up routines in athletic and clinical settings. Many previous studies reported that SS increases joint range of motion (ROM) and decreases stiffness of the muscle or muscle-tendon unit (MTU) [3,4,5,6]. Additionally, it has been reported that poor ROM and increased muscle stiffness are associated with sports-related injuries [7,8]. Increased ROM after SS is caused by a decrease in MTU stiffness and/or an increase in stretch tolerance [4,5]. In addition, changes in these factors depend on stretching intensity. Stretching intensity is often evaluated by stretching angle [9,10]. Higher intensity of stretching induces a greater decrease in MTU stiffness and greater increase in stretch tolerance [9,10]. Thus, performing stretching with higher intensity is an effective way to achieve greater improvement of ROM [11]. However, it can be difficult for practitioners to perform high intensity stretching, because high intensity stretching induces more pain.

The pain threshold can be increased during and following muscle contraction (exercise-induced hypoalgesia [EIH]) [12,13]. If practitioners are able to stretch while their pain threshold is raised, stretching intensity can be increased without an increase in pain. Thus, SS combined with muscle contraction may be an effective approach for increasing stretching intensity without increasing pain, and result in inducing a greater change of ROM and passive properties. However, it is difficult for target muscles to be stretched simultaneously with voluntary contraction, such as eccentric contraction. Electrical muscle stimulation (ES) may provide a convenient alternative to voluntary contraction. The activation of muscle fibers to produce muscle contraction is triggered by a self-generated electrical stimulus called an action potential. The same activation of muscle fibers can be triggered by artificial electrical currents by ES, provoking involuntary muscle contraction. Performing SS during muscle contraction by ES could potentially enable stretching intensity to be increased by increasing the pain threshold. Two previous studies investigated the acute and chronic effect of SS combined with ES (SS+ES) on flexibility and muscle strength [14,15]. The results revealed no significant difference in stretching intensity between the SS+ES and SS groups, although the SS+ES group exhibited a greater stretching angle in the first 4 weeks of an 8-week intervention compared with SS group [14]. The other study that examined the acute effect of SS+ES matched the stretching angle between combined and SS trials [15]. This previous study did not focus on examining the difference in stretching intensity between trials. Therefore, it is currently unclear whether SS+ES can acutely increase stretching intensity.

SS decreases muscle stiffness but not tendon stiffness [4,5]. One previous study reported that muscle stiffness was decreased after SS and proprioceptive neuromuscular facilitation stretching trials, although tendon stiffness was decreased after isometric contraction and proprioceptive neuromuscular facilitation stretching trials [16]. In addition, eccentric contraction for 6 weeks increased tendon stiffness and decreased muscle stiffness [17]. Thus, muscle stretching is needed to change muscle stiffness, whereas muscle contraction, which may relatively increase tissue loading occurring within the tendon, is needed to change tendon stiffness. SS+ES is a similar contraction pattern to eccentric contraction (i.e., muscle was stretched in length by SS while being contracted by ES), and may affect not only muscle stiffness, but also tendon stiffness. A previous study reported that both SS+ES and SS trials did not change muscle stiffness [15]. However, this previous study did not assess tendon stiffness [15]. Additionally, other passive properties, such as displacement of the muscle or tendon, would be expected to be affected differently between SS+ES and SS alone.

The purpose of the present study was to examine the effects of SS+ES on maximal dorsiflexion angle, passive torque, and displacement of muscle and tendon. The hypothesis examined in this study was that additional ES may increase stretching intensity via an increase in the pain threshold (evaluated by passive torque at maximal dorsiflexion angle in this study), subsequently enabling SS+ES to affect tendon properties. The present results may contribute to the establishment of a new stretching method in which the addition of ES to SS can induce greater stretching intensity and more significant changes in ROM, stiffness, and stretch tolerance. This new method would be helpful for athletes and/or rehabilitation patients.

## 2. Materials and Methods

### 2.1. Experimental Design

Subjects visited the laboratory four times at intervals of at least 24 h. The first visit was a familiarization trial, and the following experiment trials were involved over the subsequent three visits with random order: (a) SS+ES; (b) SS; (c) sitting at rest as a control condition (CON). During the familiarization trial, subjects practiced the passive-dorsiflexion test, SS, and ES. During the experimental trials, a passive-dorsiflexion test was performed before and after SS+ES, SS, or CON following a 5-minute warm-up on a bicycle ergometer (50 W).

### 2.2. Subjects

The number of subjects required was calculated based on the following parameters (power = 0.80, alpha = 0.05, effect size = 0.25), indicating that a minimum of 14 subjects were needed (GPower 3.1). Sixteen healthy subjects (seven male and nine female) volunteered to participate in this study (mean ± standard deviation age 20.8 ± 0.8 years, height 163.6 ± 6.4 cm, weight 53.2 ± 7.7 kg). No subjects reported a recent history of musculoskeletal injury or neuromuscular disease specific to the lower extremities. No subjects were on a structured physical training regimen. 

### 2.3. Procedures

#### 2.3.1. Passive-Dorsiflexion Test

To determine the passive torque and maximal dorsiflexion angle, each subject underwent two passive-dorsiflexion tests before and after each trial. The right foot of the subject was secured to an isokinetic machine (S-15177; Takei Scientific Instruments, Niigata, Japan) with the right knee fully extended. The seat backs were angled at 75° to the floor. In this study, the footplate angle is shown as the ankle joint angle, which is defined as 0° when the footplate is perpendicular to the floor. The dorsiflexion values are shown as positive values. The subject’s ankle joint was passively dorsiflexed at a rate of 1°/s from −30°. Dorsiflexion was stopped by pressing a switch when the subject felt discomfort in the lower limb, and the angle at that point was used as the maximum dorsiflexion angle. At the same time, the passive torque generated on the footplate was evaluated at submaximal dorsiflexion angles and the maximal dorsiflexion angle. During the passive dorsiflexion test, subjects were asked to be completely relaxed and not to offer any voluntary contraction. The measurement that recorded the greater maximum dorsiflexion angle during the two passive dorsiflexion tests was used for all subsequent analyses. Passive torque and ankle angle were converted from analogue to digital values at a sampling rate of 1.0 kHz (PowerLab 16SP; PowerLab System, AD Instruments Pty Ltd., Australia).

Passive torque at the submaximal angles was assessed at every 4° during the final 13° common to pre- and post-intervention [6,18]. MTU stiffness values were calculated as the slope of the torque-angle curve from the final 1° to 13°. Similarly, the submaximal displacement of the muscle–tendon junction (MTJ), the submaximal displacement of the tendon, muscle stiffness, and tendon stiffness were also calculated.

#### 2.3.2. Ultrasonography

The displacement of the MTJ of the medial gastrocnemius (MG) muscle was determined by B-mode ultrasonography (LOGIQ 5; GE Healthcare, Hartford, CT, USA) during the passive dorsiflexion test [4,5]. The linear array probe (12L probe; GE Healthcare) was fixed to the skin. The MTJ was visualized as a longitudinal ultrasound image and synchronized with passive torque and ankle angle output. The value relative to a reflective marker and the probe was assessed as the MTJ displacement. 

#### 2.3.3. Calculation of Tendon Displacement

A cadaveric regression model was used to estimate the change in MTU length during ankle dorsiflexion [19]. The percentage change in MTU length was calculated at every 4° during the final 13° and at the maximal dorsiflexion angle, and multiplied by the lower leg length as measured from the popliteal fossa to the lateral malleolus to estimate the MTU displacement. The tendon displacement was calculated by subtracting the displacement of MTJ from the displacement of MTU. 

#### 2.3.4. Electromyography

Electromyography (EMG) activity of the MG was measured using bipolar, disposable surface electrodes (DL-140; S&ME, Tokyo, Japan) during the passive-dorsiflexion test [4,5]. Surface electrodes were placed over the most prominent bulge of the MG. Interelectrode distance was 20 mm. EMG activity was recorded in the 5–500 Hz bandwidth with a 1.0 kHz sampling rate. The EMG amplitudes were calculated using a root mean square function for the initial and final 5° of dorsiflexion, respectively.

#### 2.3.5. Static Stretching

Subjects were secured in an isokinetic machine with the right knee fully extended and the footplate secured to the right foot. The footplate was dorsiflexed passively from −30° at a constant velocity of 1°/s [3]. The footplate was stopped by pushing a button at the position that elicited the maximal stretch sensation in the triceps surae muscle without pain. This position was held for 60 s, and then the footplate was returned to the −30°. This procedure was repeated five times without a rest [14,15]. Subjects were instructed to relax completely and not to offer any voluntary contraction.

#### 2.3.6. Electrical Stimulation

ES was conducted concurrently with SS. An electrical stimulator (Ito Espurgee; Ito Physiotherapy and Rehabilitation, Tokyo, Japan) was used to contract the MG of the right lower limb. The MG was continuously stimulated during passive dorsiflexion and 1 min of SS. One surface electrode (50 × 90 mm) was placed over the proximal MG, and the other was positioned over the distal MG. The stimulation parameters were set as an impulse frequency of 70 Hz and an impulse width of 300 μs [14,15,20]. Current intensity (mA) was first regulated to the maximal tolerable amperage without pain at 30° of plantar flexion position at the first set of SS. After that, subjects regulated the current intensity to the maximal tolerable amperage without pain during passive dorsiflexion and subsequent 1 min of SS. After SS for 1 min, the ES was interrupted until the ankle was returned to the plantar flexion position. Subjects were then asked to report the highest current intensity which they were able to reach during 1 min of SS. After the second set, subjects regulated current intensity to half of the highest current intensity that subjects were able to reach in the previous set at 30° of plantar flexion position. Thereafter, passive dorsiflexion was started, and subjects were instructed to freely regulate the current intensity as mentioned above.

### 2.4. Data Reliability

The test–retest reliability for maximal dorsiflexion angle, displacement of the MTJ at the maximal dorsiflexion angle, and passive torque at maximal dorsiflexion angle were reported in previous studies [15,21].

### 2.5. Statistical Analyses

SPSS version 22.0 (SPSS, Inc., Chicago, IL, USA) was used to conduct statistical analyses. The Shapiro–Wilk test was used to screen for normal distribution. The assumption of sphericity was used to assess homogeneity of the variance assumption. The Greenhouse–Geisser or Huynh–Feldt adjustment was used to correct violation when parameters that did not meet the assumption of sphericity. A three-way analysis of variance (ANOVA; time [pre- or post-intervention] × trial [SS+ES, SS, or CON] × angle [final 1°, final 5°, final 9°, final 13°, or maximal dorsiflexion angle]) was used to analyze the passive torque, the displacement of MTJ, and the displacement of the tendon. A three-way analysis of variance (ANOVA; time [pre- or post-intervention] × trial [SS+ES, SS, or CON] × angle [final 1°, final 5°, final 9°, final 13°]) was used to analyze the stiffness of MTU, the stiffness of the muscle, and the stiffness of the tendon. A three-way ANOVA (time [pre- or post-intervention] × trial [SS+ES, SS, or CON] × portion [initial 5°, or final 5°]) was used to analyze the EMG amplitudes of the MG. A two-way ANOVA (time [pre- or post-intervention] × trial [SS+ES, SS, or CON]) was used to analyze the maximal dorsiflexion angle. A two-way ANOVA (set [1st, 2nd, 3rd, 4th, or 5th set] × trial [SS+ES, SS]) was used to analyze the angle of SS. A one-way ANOVA (trial [SS+ES, SS, or CON]) was used to analyze the difference in maximal dorsiflexion angle from pre- to post-intervention. Follow-up analyses were performed using lower-order ANOVA and t-tests with Bonferroni correction. Statistical significance was set at *p* ≤ 0.05. All data are reported as means ± standard deviation.

## 3. Results

### 3.1. Angle of SS

There was no significant two-way interaction between set and trial (*p* = 0.319, ηp2 = 0.074), and no significant main effect for trial (*p* = 0.835, ηp2 = 0.003). There was a significant main effect of set (*p* = 0.037, ηp2 = 0.229). However, post hoc testing revealed that there was no significant difference between sets (all *p* > 0.05).

### 3.2. Maximal Dorsiflexion Angle

There was a significant two-way interaction between trial and time (*p* = 0.021, ηp2 = 0.261). Post hoc testing revealed that maximal dorsiflexion angle was significantly increased after SS+ES (*p* < 0.001) and SS (*p* = 0.001), but not after CON (*p* = 0.081) (Figure 1a).

### 3.3. Difference in Maximal Dorsiflexion Angle

There was a significant one-way interaction among trials (*p* = 0.021, ηp2 = 0.261). Post hoc testing revealed that the difference in maximal dorsiflexion angle after SS+ES was significantly greater than that after CON (*p* = 0.026) (Figure 1b). 

### 3.4. Passive Torque

There was a significant three-way interaction between trial, time, and angle (*p* = 0.041, ηp2 = 0.204). Post hoc testing revealed that passive torque at maximal dorsiflexion angle was increased after all trials (SS+ES: *p* = 0.001, SS: *p* = 0.005, CON: *p* = 0.008). In addition, passive torque at final 1° (*p* = 0.028), 5° (*p* = 0.025), and 9° (*p* = 0.032) was decreased after SS+ES, and passive torque at final 1° (*p* = 0.021), and 5° (*p* = 0.043), was increased after CON. Additionally, passive torque was increased with an increase in angle (all *p* < 0.05), except for between final 13° and the maximal dorsiflexion angle at pre-intervention in all trials (all *p* > 0.05), and between final 13° and the maximal dorsiflexion angle at post-intervention in the CON condition (*p* = 0.079), (Table 1).

### 3.5. Displacement of MTJ and Tendon

There was no significant three-way interaction between trial, time, and angle (*p* = 0.338, ηp2 = 0.072), and no significant two-way interaction between trial and time (*p* = 0.564, ηp2 = 0.030), or trial and angle (*p* = 0.636, ηp2 = 0.035) for displacement of MTJ. However, there was a significant two-way interaction between time and angle (*p* = 0.001, ηp2 = 0.390). Post hoc testing revealed that the displacement of MTJ at maximal dorsiflexion angle was increased after all trials (*p* = 0.011,). In addition, the displacement of MTJ was increased with increase in angle (*p <* 0.001) except for between final 13° and maximal dorsiflexion angle at pre-intervention (*p* = 0.088) (Table 2).

There was a significant three-way interaction between trial, time and angle (*p* = 0.040, ηp2 = 0.141) for the displacement of the tendon. Post hoc testing revealed that the displacement of the tendon at maximal dorsiflexion angle was increased after SS+ES (*p* < 0.001) and SS (*p* = 0.023). In addition, the displacement of the tendon at maximal dorsiflexion angle in the post-intervention value was significantly greater for ES+SS than that for SS (*p* = 0.030), but was not significantly different for CON (*p* = 0.053). Additionally, displacement of the tendon was increased with increase in angle (*p <* 0.001) except for between final 13° and the maximal dorsiflexion angle at pre-intervention in all trials (*p >* 0.05), and between final 13° and the maximal dorsiflexion angle at post-intervention in the CON condition (*p =* 0.126) (Table 3).

### 3.6. Stiffness of MTU, Muscle, and Tendon

There was no significant three-way interaction between trial, time, and angle (*p* = 0.397, ηp2 = 0.058), and no significant two-way interaction between trial and time (*p* = 0.216, ηp2 = 0.098), or trial and angle (*p* = 0.470, ηp2 = 0.043), or time and angle (*p* = 0.163, ηp2 = 0.126) for stiffness of MTU. In addition, there were no significant main effects of trial (*p* = 0.252, ηp2 = 0.088) and time (*p* = 0.859, ηp2 = 0.002), but there was a significant main effect of angle (*p* < 0.001, ηp2 = 0.726). Post hoc testing revealed that stiffness of MTU was increased with increase in angle (all *p* < 0.001) (Table 4).

There was no significant three-way interaction between trial, time, and angle (*p* = 0.959, ηp2 = 0.002), and no significant two-way interaction between trial and time (*p* = 0.421, ηp2 = 0.041), or trial and angle (*p* = 0.498, ηp2 = 0.032), or time and angle (*p* = 0.258, ηp2 = 0.071) for stiffness of muscle. In addition, there were no significant main effects of trial (*p* = 0.286, ηp2 = 0.065) and time (*p* = 0.780, ηp2 = 0.004), but there was a significant main effect of angle (*p* = 0.001, ηp2 = 0.481). Post hoc testing revealed that stiffness of muscle was increased with increase in angle (all *p* < 0.05).

There was no significant three-way interaction between trial, time, and angle (*p* = 0.461, ηp2 = 0.050), and no significant two-way interaction between trial and time (*p* = 0.671, ηp2 = 0.026), or trial and angle (*p* = 0.484, ηp2 = 0.048), or time and angle (*p* = 0.498, ηp2 = 0.048) for stiffness of tendon. In addition, there were no significant main effects of trial (*p* = 0.296, ηp2 = 0.076) and time (*p* = 0.380, ηp2 = 0.052), but there was a significant main effect of angle (*p* = 0.001, ηp2 = 0.512). Post hoc testing revealed that stiffness of the tendon was increased with an increase in angle (all *p* < 0.05) except for between final 9° and final 13° (*p* = 0.348).

### 3.7. EMG

There were no significant three-way interactions between trial, time, and portion (*p* = 0.432, ηp2 = 0.050), and no significant two-way interactions for trial and time (*p* = 0.431, ηp2 = 0.051), time and portion (*p* = 0.135, ηp2 = 0.143), or trial and portion (*p* = 0.591, ηp2 = 0.022). In addition, no significant main effects were detected for trial (*p* = 0.535, ηp2 = 0.035), time (*p* = 0.905, ηp2 = 0.001), or portion (*p* = 0.771, ηp2 = 0.006) (Table 5).

## 4. Discussion

The purpose of this study was to examine the effects of SS+ES on maximal dorsiflexion angle, passive torque, and displacement of muscle and tendon. The main finding of this study was that the difference in maximal dorsiflexion angle from pre- to post-intervention after SS+ES trials was significantly greater than that after CON trials, whereas there was no significant difference in maximal dorsiflexion angle after SS trials compared with that after CON trials. Passive torque at submaximal angle was significantly decreased after SS+ES trials, but not after SS trials. In addition, displacement of tendon at maximal dorsiflexion angle at post-intervention was greater after SS+ES than SS trials. Thus, SS+ES exerted an additional effect on maximal dorsiflexion angle, passive torque, and displacement of tendon compared with SS alone.

The present study clarified that ES combined with SS is effective for increasing the maximal dorsiflexion angle. The maximal dorsiflexion angle was increased after both SS+ES and SS trials in this study. In addition, the difference in maximal dorsiflexion angle from pre- to post-intervention after SS+ES trials was greater than that in the CON condition, whereas there was no significant difference between the SS and CON condition. The increase in maximal dorsiflexion angle after SS trials in the present study was similar to the findings of a previous study in which SS was performed for 5 min, but the increase in maximal dorsiflexion angle after SS+ES trials was greater [3]. Thus, the current findings suggested that SS+ES induced a greater increase in maximal dorsiflexion angle than SS alone. The increased maximal dorsiflexion angle for SS+ES was caused by a decrease in submaximal passive torque and an increase in stretch tolerance, whereas increased maximal dorsiflexion angle for SS was caused by an increase in stretch tolerance alone. However, a previous study reported that SS+ES and SS trials exhibited a similar increase in maximal dorsiflexion angle [15]. The discrepancy between the current results and those of this previous study could be explained by a difference in the SS method used. In the current study, SS angle was modified by subjects in every set, whereas SS angle in the previous study was controlled by the investigator in all sets, and matched between trials [15]. Additionally, ES was administered during plantarflexion movement and maximal dorsiflexion position as SS in the current study, whereas ES was administered only during the maximal dorsiflexion position in the previous study [15]. Thus, it is possible that the target muscle was contracted in a similar way to eccentric contraction, in this study, whereas it was contracted in a similar way to isometric contraction in the previous study. This difference may have induced differences in tissue strain despite the same SS angle.

The most interesting finding in the current study was the greater increase in displacement of the tendon at maximal dorsiflexion angle after SS+ES trials. It was previously reported that SS does not affect tendon properties, but tendon stiffness was decreased after muscle voluntary contraction, such as isometric contraction, proprioceptive neuromuscular facilitation stretching, and eccentric contraction [4,5,16,17]. Because the maximal displacement of the tendon after SS+ES was increased in the current study, ES may have an effect on the tendon that is similar to voluntary muscle contraction and become an alternative method for voluntary contraction. 

SS angle was not increased after additional ES compared with SS alone. A previous study reported that the pressure pain threshold at the quadriceps muscle was increased during 30% of maximal voluntary contraction of the muscle [13]. Another study reported that the pressure pain threshold at the index finger was increased after maximal velocity concentric contractions of the elbow flexors [12]. This kind of decrease in pain perception during and following acute voluntary muscle contraction was named EIH. Thus, it is speculated that ES would also induce EIH-like voluntary contraction, resulting in an increase in maximal dorsiflexion angle during SS. However, the results revealed no significant difference in SS angle between ES+SS and SS trials. This finding may have been caused by differences in the definition of SS angle. Because this study defined the SS angle as the dorsiflexion position that provoked the sensation of maximal stretch in the triceps surae muscle without pain, the SS angle was not synonymous with the pain threshold. When using EIH for SS intensity, it is important to perform SS with high intensity that is maximally tolerable despite some pain. In that case, additional ES may allow SS to be performed with a greater angle, or to be performed at a given angle with less pain.

It is currently unclear why passive torque was decreased at submaximal angles only after SS+ES. The current results revealed that SS+ES decreased the passive torque at submaximal angles and increased stretch tolerance, whereas SS increased stretch tolerance alone. Thus, the decrease in passive torque at submaximal angles appears to be an additional effect caused by ES. Several hypotheses have been proposed to explain the mechanisms responsible for the stretching-induced decreases in passive torque at submaximal angles, including increases in tendon compliance [22,23], increases in fascicle length and muscle compliance [3,4,24], and alterations in connective tissues [25]. However, the current findings revealed no significant differences in stiffness of the MTU, muscle, or tendon. In addition, there was no significant difference in the displacement of muscle and tendon at submaximal angles. Therefore, decreased passive torque at submaximal angles may have been caused by alterations in connective tissues, which include the noncontractile muscle proteins such as titin, and desmin [6,25,26]. Herda et al. [27] reported that SS with a constant angle decreased passive torque but not MTU stiffness, whereas SS with constant torque decreased passive torque and MTU stiffness. Changes in MTU stiffness (i.e., changes in the slope of the torque-angle curve) may reflect changes in the viscoelastic properties of MTU. Gajdosik [25] reported that SS with a constant angle only affects viscosity. Additionally, Herda et al. [27] suggested that SS with a constant angle may affect only the viscosity but not the elasticity, although SS with a constant angle may affect both viscosity and elasticity. However, it has been reported that the rate of decrease in passive torque at a submaximal angle is affected by stretching intensity [10]. Although there was no significant difference in SS angles between trials, stretching intensity to the MTU in SS+ES trials may be greater than that in SS trials because of ES-induced muscle contraction. However, because passive torque during SS was not measured in this study, this study cannot reveal whether this possibility is correct or not, and further is needed.

ES+SS appears to be one of the most effective exercises to increase ROM. In the present study, one bout of SS+ES increased the maximal dorsiflexion angle (Pre to Post-change: 5.3°, effect size: 0.73). However, previous studies have reported that ROM did not change after a single bout of aerobic exercise [28] or chronic resistance training [29]; McNair [28] reported no significant change in maximal dorsiflexion angle after aerobic exercise (10 minutes of jogging) (Pre to Post-change: 0.2°, effect size: 0.03). It also reported no significant difference in knee flexion ROM after 5 weeks of resistance training (Pre to Post-change: 0.6°, effect size: 0.15) [29]. On the other hand, ROM has been reported to increase after chronic interval training [30], yoga [31], and Pilates training [32]; Batrakoulis et al. [30] reported an increase in ankle angle (Pre to Post-change: 4.3°, effect size: 1.30) after 1 year of interval training. A recent review demonstrated that chronic yoga increased lower extremity flexibility (primarily sit-and-reach test) (effect size: 0.488) [31]. In addition, 6 weeks of Pilates training improved the sit-and-reach test (pre- to post-change: 8.0 cm, effect size: 2.89) [32]. Comparison of the sit-and-reach test and goniometric measurement is difficult because the sit-and-reach test contributes to multi-joint angles, whereas the goniometric measurement contributes to single-joint angles. Thus, it was unclear whether SS+ES or yoga, or Pilates were more effective in increasing ROM. In addition, it is obviously impossible to directly compare the results of the acute and chronic periods. However, a previous study examining the chronic effects of SS+ES at an intensity and duration similar to that of the present study reported that SS+ES for 8 weeks increased the maximum dorsiflexion angle (Pre to Post-change: 7.4°, effect size: 1.07). Thus, SS+ES conducted in this study may be as effective as or more effective than a single bout of effect when conducted over long-term intervention. Therefore, when the amount of pre- and post-differences and the effect size are comprehensively compared, the ROM-increasing effect of SS+ES is expected to be equal to or greater than that of aerobic exercise, resistance training, and interval training.

The strength (novelty) of this study is SS combined with ES. When combining SS with voluntary contraction such as eccentric contractions or proprioceptive neuromuscular facilitation stretching, partner and training equipment are needed. It also carries the risk of fatigue and injury due to maximum voluntary muscle contraction. On the other hand, if SS is combined with ES, self-implementation is possible. Additionally, ES can mobilize more muscle fibers with relatively low effort [33], which may reduce the risk of fatigue and injury.

There are some limitations in this study. First, sex difference was not examined in this study. Previous studies investigated that passive poperies such as displacement of MTJ and stiffness of muscle were different between males and females [34]. In addition, it is also reported that the acute effects of static stretching on ROM and stretch tolerance were different between males and females [35]. Thus, the combined effects of SS and ES on ROM and passive properties may also have gender differences. Second, the experimental site for this study was the lower leg only. Future studies are needed to determine the effects on other sites.

## 5. Conclusions

The difference from pre- to post-intervention in maximal dorsiflexion angle in SS+ES trials was greater compared with that in the CON condition. However, there were no significant differences in the difference from pre- to post-intervention in maximal dorsiflexion angle between SS and SS+ES, and SS and CON conditions. Passive torque at submaximal dorsiflexion angle was significantly decreased only after SS+ES trials. Tendon displacement at maximal dorsiflexion angle was higher after SS+ES trials compared with SS trials. These results revealed additional effects of adding ES to SS on maximal dorsiflexion angle, passive torque, and tendon displacement. Therefore, combined SS and ES exercise may be a new alternative to traditional SS for those who want to increase flexibility. For athletes in disciplines where flexibility is an important factor in athletic performance (such as ballet dancer and gymnasts) and for rehabilitation patients aiming to improve flexibility, combined SS and ES exercises can be an effective means of altering ROM and passive properties.

## Figures and Tables

**Figure 1 sports-11-00010-f001:**
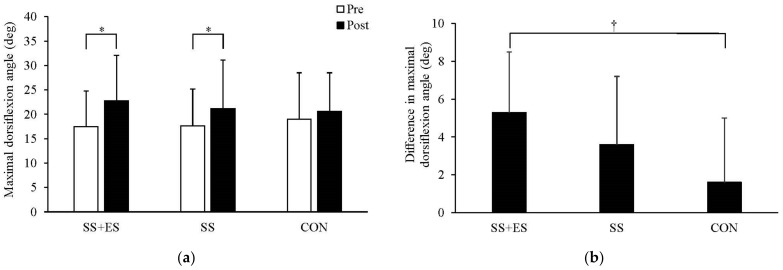
Effect of combined static stretching (SS) and electrical stimulation (ES), SS, and control (CON) on maximal dorsiflexion angle. (**a**) Absolute values of maximal dorsiflexion angle at pre- and post-intervention. (**b**) Differences in maximal dorsiflexion angle from pre- to post-intervention. * Significantly different from the pre-intervention value. † Significantly different from the CON condition.

**Table 1 sports-11-00010-t001:** Passive torque (Nm) during the final 13° before and after SS+ES, SS, and CON.

		Final1°	Final5° ^†^	Final9° ^††^	Final13° ^†††^	End ROM ^††††^
SS+ES	Pre	7.1 ± 3.3	9.1 ± 4.3	11.7 ± 5.3	14.9 ± 6.9	14.9 ± 6.9
	Post	6.7 ± 3.3 *	8.6 ± 4.2 *	11.1 ± 5.2 *	14.3 ± 6.5	19.4 ± 9.9 *
SS	Pre	7.0 ± 4.0	9.0 ± 5.1	11.6 ± 6.4	14.7 ± 8.0	14.7 ± 8.0
	Post	6.8 ± 3.8	8.8 ± 4.8	11.3 ± 6.0	14.4 ± 7.4	17.6 ± 10.0 *
CON	Pre	7.7 ± 4.3	9.9 ± 5.5	12.6 ± 6.9	15.9 ± 8.3	16.3 ± 8.4
	Post	8.1 ± 4.5 *	10.4 ± 5.8 *	13.2 ± 7.1	16.6 ± 8.6	19.0 ± 10.0 *

* *p* < 0.05 vs. pre-intervention value at the same angle in the same condition; ^†^
*p* < 0.05 vs. final1°; ^††^
*p* < 0.05 vs. final1° and final5°, ^†††^
*p* < 0.05 vs. final1°, final5°, and final9°, ^††††^
*p* < 0.05 vs. final1°, final5°, final9°, and final13° (except for pre-intervention in all trials, and post-intervention in the CON); Values represent means ± SD.

**Table 2 sports-11-00010-t002:** Displacement of muscle-tendon junction (mm) during the final 13° before and after SS+ES, SS, and CON.

		Final1°	Final5° ^†^	Final9° ^††^	Final13° ^†††^	End ROM ^††††^
SS+ES	Pre	0.0 ± 0.0	1.5 ± 0.7	2.8 ± 1.0	3.9 ± 1.3	3.9 ± 1.2
	Post	0.0 ± 0.0	1.3 ± 0.8	2.4 ± 0.8	3.6 ± 1.4	4.6 ± 1.9 *
SS	Pre	0.0 ± 0.0	1.6 ± 0.7	3.1 ± 1.0	4.0 ± 1.5	4.1 ± 1.4
	Post	0.0 ± 0.0	1.5 ± 0.6	2.8 ± 0.9	4.0 ± 1.0	5.2 ± 1.7 *
CON	Pre	0.0 ± 0.0	1.3 ± 0.8	2.7 ± 0.8	3.8 ± 1.2	4.2 ± 1.3
	Post	0.0 ± 0.0	1.3 ± 0.6	2.8 ± 1.1	4.0 ± 1.2	4.6 ± 1.8 *

* *p* < 0.05 vs. Pre-intervention value at the same angle in the same condition, ^†^
*p* < 0.05 vs. final1°, ^††^
*p* < 0.05 vs. final1° and final5°, ^†††^
*p* < 0.05 vs. final1°, final5°, and final9°, ^††††^
*p* < 0.05 vs. final1°, final5°, final9°, and final13° (except for pre-intervention); Values represent means ± SD.

**Table 3 sports-11-00010-t003:** Displacement of tendon (mm) during the final 13° before and after SS+ES, SS, and CON.

		Final1°	Final5° ^†^	Final9° ^††^	Final13° ^†††^	End ROM ^††††^
SS+ES	Pre	0.0 ± 0.0	1.3 ± 0.6	2.6 ± 0.9	4.1 ± 1.1	4.1 ± 1.1
	Post	0.0 ± 0.0	1.5 ± 0.7	3.1 ± 0.7	4.5 ± 1.0	6.6 ± 2.0 *^,§^
SS	Pre	0.0 ± 0.0	1.2 ± 0.6	2.4 ± 0.8	4.0 ± 1.2	4.1 ± 1.3
	Post	0.0 ± 0.0	1.3 ± 0.5	2.7 ± 0.6	4.1 ± 0.6	5.1 ± 1.5 *
CON	Pre	0.0 ± 0.0	1.4 ± 0.7	2.7 ± 0.7	4.2 ± 1.0	4.2 ± 1.1
	Post	0.0 ± 0.0	1.4 ± 0.5	2.6 ± 0.8	4.0 ± 0.7	5.0 ± 1.7

* *p* < 0.05 vs. Pre-intervention value at the same angle in the same condition, ^§^
*p* < 0.05 vs. Post-intervention value at the same angle in the SS condition, ^†^
*p* < 0.05 vs. final1°, ^††^
*p* < 0.05 vs. final1° and final5°, ^†††^
*p* < 0.05 vs. final1°, final5°, and final9°, ^††††^
*p* < 0.05 vs. final1°, final5°, final9°, and final13° (except for pre-intervention in all trials, and post-intervention in the CON); Values represent means ± SD.

**Table 4 sports-11-00010-t004:** Stiffness of muscle-tendon unit (Nm/°) during the final 13° before and after SS+ES, SS, and CON.

		Final1°	Final5° ^†^	Final9° ^††^	Final13° ^†††^
SS+ES	Pre	0.42 ± 0.21	0.58 ± 0.27	0.73 ± 0.34	0.89 ± 0.39
	Post	0.39 ± 0.21	0.55 ± 0.25	0.71 ± 0.39	0.87 ± 0.39
SS	Pre	0.45 ± 0.26	0.58 ± 0.31	0.71 ± 0.39	0.83 ± 0.49
	Post	0.41 ± 0.24	0.56 ± 0.28	0.71 ± 0.35	0.86 ± 0.44
CON	Pre	0.48 ± 0.31	0.61 ± 0.32	0.75 ± 0.38	0.88 ± 0.46
	Post	0.50 ± 0.31	0.64 ± 0.34	0.78 ± 0.39	0.91 ± 0.47

Stiffness of muscle-tendon unit was increased with increases in angle. ^†^
*p* < 0.05 vs. final1°, ^††^
*p* < 0.05 vs. final1° and final5°, ^†††^
*p* < 0.05 vs. final1°, final5°, and final9°; Values represent means ± SD.

**Table 5 sports-11-00010-t005:** EMG values (mV) at initial during the initial 5° and final 5° during passive dorsiflexion test before and after SS+ES, SS, and CON.

		Initial 5°	Final 5°
SS+ES	Pre	0.040 ± 0.010	0.039 ± 0.013
	Post	0.040 ± 0.011	0.043 ± 0.004
SS	Pre	0.042 ± 0.004	0.041 ± 0.009
	Post	0.041 ± 0.003	0.040 ± 0.003
CON	Pre	0.042 ± 0.003	0.040 ± 0.009
	Post	0.038 ± 0.011	0.039 ± 0.011

Values represent means ± SD.

## Data Availability

Data relating to this article will be made available upon request to the corresponding author.

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
