# Peer review of "Combined Static Stretching and Electrical Muscle Stimulation Induce Greater Changes in Range of Motion, Passive Torque, and Tendon Displacement Compared with Static Stretching"

_sports, 2023, doi:10.3390/sports11010010_

Round 1

Reviewer 1 Report

First of all, I must say that I read the article line by line with interest and excitement. In my opinion, the article needs major changes and corrections before it can be accepted for possible publication. Below are my recommendations that I think will make the article more readable and scientifically qualified.

In the abstract: all p values should be added. Readers may wish to understand whole paper via looking abstract.

Line 50-56: here is sound little confusing, is SS+ES not known for its ability to increase stretching intensity ? however, in the above, line 51, you are saying SS+ES did not affect stretching intensity compared to SS only group? Clarify these sentences.

Line 57: why SS dont decrease tendon stifness, what mechanisms is responsible for this occurrence?

Line 57-59: Combined SS and PNF decreased muscle stifness but Combined isometric contraction and PNF decreased tendon stifness ? here is complicated, check the meaning and revise the sentences.

İn the whole intro, it is not explained scientificly why this study conducted and what this study contributes to the literature, I am not convinced about the necessity of this study’s application. Author should clarify the whole intro scientificly and add SS and ES mechanisms of action.

Material method

How did you perform randomization ?

How did you calculate sample size and via which software ? which study was used to calculate power ?

Method 2.3.2. ultrasonography

Need a citation.

2.3.3. Calculation of tendon displacement

Need a citation, you created this methods or you use this methods because someones created it before ?

2.3.4. Electromyography

Cite your reference

2.3.5. Static stretching

Cite your reference

2.4. Data reliability

Using “I” does not sounds scientificly. Author should use terms such as “It was observed …..” or “It has been already reported that….”

Results 3.1:

İnclude p values before effect size values.

Throughout the results section, add all p values for reference results.

Why author did not present EMG values ? I think it should be provided.

Line 319-322: revise this sentence, it can be shorten. Sounds repetition.

Line 337: I think, it is not convinient to use   “I”, maket the sentence passive and go on.

Add a limitations paragragh before conclusion.

Erase the line 374-375, no need to replicate the “aim” that mentioned already at the beginning of the discussion section.

Summarize the results in the line 374-382

Give a clue or recommendation to the athletes or coaches or whoever use this information ?

Reviewer 2 Report

I suggest to clarify with significance symbols were there are differences at positions within conditions. The read needs to read to text or the legend of the Table where significant differences occur.

Ls 33-39. I suggest to clarify what is meant by and how stretching intensity is quantified. Stretching intensity is mentioned in the hypothesis at the end of the abstract so some clarification is welcomed.

L63. Motor unit recruitment during an eccentric contraction is likely different than recruitment during static stretching and electrical stimulation so to suggest similarity needs to be clarified/revised. See also Ls 319-320.

L72. In the hypothesis is mention of an increase in the pain threshold but there are no measurements to support or not support the hypothesis.

L84. Change “samples” to “subjects”.

L168. I suggest to delete “I have already reported the” and replace with “The”.

L169. I suggest to replace “in previous studies” to “ were reported in previous studies”.

Ls286-288. Please delete “This section……drawn”.

Ls 302-304. I suggest clarify “In addition, the difference in maximal dorsiflexion angle from pre- to post-intervention after SS+ES trials was greater than that in the CON condition, whereas there was no significant difference between the SS and SS+ES conditions, or CON condition.” There seems to a contradiction in this statement.

L359. Perimysium is not a non-contractile protein. Please revise.

Ls 370-371. Please clarify which study you are referring to.

L377. Please clarify that “but the SS condition exhibited no significant difference”, that there was no difference compared to the other conditions, as in the SS condition, there was an increase pre-post for maximal dorsiflexion angle.

Reviewer 3 Report

General comments

 The author has clearly stated that the purpose of the study was to examine determine the combined effects of static stretching and electrical muscle stimulation on maximal dorsiflexion angle and passive properties. The paper is well-written, easy to follow and adds merit to the vital role of stretching in health and performance. Given this approach, this work can enhance future attempts in similar research area. However, I have highlighted a few suggestions and concerns in my specific comments section (below) that need to be addressed before considering whether this work should be published or not.

Specific comments

 ABSTRACT

 - Exact p values, % of change and effect sizes should be presented in results.

 INTRODUCTION

 - I suggest adding a sentence about the beneficial role of flexibility training in general health of adults according to the latest guidelines by the World Health Organization (1) and the American College of Sports Medicine (2).

- A statement about the popularity of the electrical muscle stimulation and flexibility training in the health and fitness industry worldwide according to the latest report published by the American College of Sports Medicine (3), it could be a useful addition.

 Suggested References:

1.      Bull FC, Al-Ansari SS, Biddle S, Borodulin K, Bumanat MP, Cardonal G, et al. World Health Organization 2020 guidelines on physical activity and sedentary behaviour. Br J Sports Med 2020; 54(24): 1451-1462.

2.      American College of Sports Medicine; Liguori, G.; Feito, Y.; Fountaine, C.; Roy, B.A. ACSM’s Guidelines for Exercise Testing and Prescription, 11th ed.; Wolters Kluwer Health: Philadelphia, PA, USA, 2021.

3.      Kercher VM, Kercher K, Bennion T, Levy P, Alexander C, Amaral PC, et al. 2022 Fitness Trends from Around the Globe. ACSMs Health Fit J 2022; 26(1): 21-37.

 MATERIALS AND METHODS

 -          Was this study registered as a clinical trial before the commencement of the intervention? If so, you should mention that. Otherwise, explain why you did not register this study in advance at any national or international database in order to raise the credibility and transparency of your work.

  DISCUSSION

 - Discuss the results observed for other exercise interventions such as aerobic training, resistance training, high-intensity interval training (4), yoga (5), and Pilates (6) in various musculoskeletal fitness parameters such as passive range of motion and flexibility in various joints, including the ankle.

 - Strengths (i.e., novelty) and limitations (i.e., small sample size, one assessed joint and movement, etc.) should be presented in a separate paragraph at the end of the discussion section.

- In conclusions, you should underline the main findings and suggest future research attempts in this area while highlighting potential practical implications.

 Suggested reference:

4.      Batrakoulis, A.; Jamurtas, A.Z.; Tsimeas, P.; Poulios, A.; Perivoliotis, K.; Syrou, N.; Papanikolaou, K.; Draganidis, D.; Deli, C.K.; Metsios, G.S.; et al. Hybrid-type, multicomponent interval training upregulates musculoskeletal fitness of adults with overweight and obesity in a volume-dependent manner: A 1-year dose-response randomised controlled trial. Eur. J. Sport Sci. 2022, online ahead of print. https://doi.org/10.1080/17461391.2021.2025434.

5.      Shin, S. Meta-Analysis of the Effect of Yoga Practice on Physical Fitness in the Elderly. Int. J. Environ. Res. Public Health 2021, 18, 11663

6.      Campos, R.R.; Dias, J.M., Pereira, L.M.; Obara, K.; Barreto, M.S.; Silva, M.F.; et al. Effect of the Pilates method on physical conditioning of healthy subjects: a systematic review and meta-analysis. J Sports Med Phys Fitness 2016, 56, 864–873.

Round 2

Reviewer 1 Report

I thank the authors for making recommendations to improve the quality of this article. I think the article looks better now.

Reviewer 2 Report

Thank your for the considered response to my suggestions and comments.